# Experimental Analysis of Tribological Processes in Friction Pairs with Laser Borided Elements Lubricated with Engine Oils

**DOI:** 10.3390/ma13245810

**Published:** 2020-12-19

**Authors:** Janusz Lubas, Wojciech Szczypiński-Sala, Paweł Woś, Edyta Zielińska, Krzysztof Miernik

**Affiliations:** 1Faculty of Mechanical Engineering and Aeronautic, Rzeszow University of Technology, Powstańców Warszawy 8, 35-959 Rzeszów, Poland; pwos@prz.edu.pl (P.W.); ezielins@prz.edu.pl (E.Z.); 2Faculty of Mechanical Engineering, Cracow University of Technology, al. Jana Pawła II 37, 31-864 Cracow, Poland; ws@mech.pk.edu.pl; 3Faculty of Materials Engineering and Physics, Cracow University of Technology, al. Jana Pawła II 37, 31-864 Cracow, Poland; kmiernik@pk.edu.pl

**Keywords:** laser boriding, engine oil, wear, friction, surface layer

## Abstract

The present study discusses the influence of engine oils on the tribological parameters of sliding couples with laser borided surface layer. The borided layer was formed on specimens made from AISI 5045 steel by laser remelting of a surface layer coated with amorphous boron. The sliding friction and wear process was carried out on the pairs with AISI 5045 steel and SAE-48 bearing alloys which were lubricated with 5W-40 and 15W-40 engine oils. The investigation showed significant differences in the friction coefficient and temperature in the tested pairs with the laser borided surface layer. In the couples lubricated with 5W-40 engine oil, the tested parameter of friction was higher than in the couples lubricated with 15W-40 engine oil. The couples lubricated with 5W-40 engine oil showed more intensive wear of SAE-48 bearing alloy in contact with the laser borided surface layer than the pairs lubricated with 15W-40 engine oil. The laser borided surface layer used in friction pairs leads to the destruction of the lubricating properties of engine oils and reduces its resistance to scuffing.

## 1. Introduction

The structural elements of machines are subjected to various unfavorable operating factors that shorten their service life, which, in turn, affects the reliability of the entire structure. The friction pairs used in internal combustion engines are exposed to an intensive destruction processes caused by the abrasive wear, adhesion, oxidation and cavitation processes. The intensity of these processes is influenced by the structure of the friction pair, materials of the friction elements, surface treatment, loading and lubrication conditions.

In the currently used constructions of friction pairs, friction and wear are the result of appropriate shaping of the contacting surface layers. In the case of most components operating under frictional conditions, a surface layer of high hardness and wear resistance is required. Sometimes, they also require increased fatigue resistance, corrosion resistance, heat resistance and creep resistance. One of the surface treatment methods which enables the formation of the required structure of the surface layer of constructional elements operating under friction conditions is laser treatment [1,2,3,4,5,6]. The surface layers produced in this way show favorable tribological properties in the form of the required hardness, fracture toughness, abrasion resistance and corrosion resistance [7,8,9,10,11,12,13,14,15]. Improving the microstructure of surface layers and coatings is also mentioned as one of the advantages of using laser techniques in surface treatment [14,16,17]. The study of wear of the laser-treated steel shows lower values of friction coefficients and lower wear of samples [18,19].

One of the elements that can be used in the laser remelting process is boron. This element forms stable and hard phases with iron, and laser treatment allows to eliminate the textured, coniferous structure of the surface layer [2,14,20,21]. Other authors indicate that this process also provides a lower coefficient of friction, good oxidation and resistance to erosion [14,20,22]. Boron content in the laser treatment steel layer significantly reduces the wear under dry friction conditions [15,22,23,24]. Tests in dry sand/rubber wheel conditions showed that the abrasive wear of Fe_50_Mn_30_Co_10_Cr_10_ was reduced by up to 30% compared to the material without laser treatment [22]. The results of EN25 steel tests showed that laser boriding increases corrosion resistance and causes uniform corrosion of the surface of the test sample, which results from the formation of iron borides during laser processing [20,22]. However, some authors indicate that replacing diffusion drilling in the case of elements requiring high fatigue strength by laser drilling is not recommended [24].

The surface cracks occurring after laser remelting are then the cause of the fatigue crack. In the case of laser borided carbon steels, the carbon content influences the heating and cooling of the material. It has been shown that higher concentration of carbon in the laser borided steel influences the increase in peak temperature and cooling rate, and also increases the depths of re-melted zone and heat-affected zone [25]. The process of boriding nickel alloys is difficult when using classical methods or requires the use of proprietary boriding agents [26]. The use of the laser beam allows the melting of the boron layer and the creation of a boron zone consisting of iron, nickel and chromium borides. The hardness of this layer is comparable to that obtained in the case of diffusion drilling, with its much greater thickness (346 or 467 µm depending on the power of the laser used) [27,28,29]. Laser boriding of nickel alloy increases its wear resistance, which may be ten times higher compared to the untreated material [26,27,29].

Although real components usually work in a mixed lubrication mode [30], most publications limit the scope of their research to abrasive wear [15,19,21,22]. Garcia et al. [31] showed that partially laser remelted surfaces of plasma spray coatings reduce wear only under certain pressures and sliding velocities. Moreover, they observed that higher percentage of the remelted surface may cause inadequate lubrication conditions and increase the wear of elements [31,32]. Most friction pairs require lubrication, it is important to determine the influence of laser remelting for tribological processes under limited lubrication conditions.

The aim of the present study is to investigate the influence of the selected classes of engine oils on friction and wear processes in friction pairs containing elements with laser borided surface layer. The use of boriding causes the formation of hard borides, which are resistant to wear, and the use of laser boriding assures that surface layer does not show a tendency to crack. Such surface layers can be used in friction pairs in internal combustion engines lubricated with 5W-40 and 15W-40 engine oils. This requires an answer to the question of what is the influence of engine oils on tribological processes under limited lubrication conditions. This paper also describes the anti-seizure parameters of the engine oils used for lubrication of the sliding pairs with elements containing boron.

## 2. Materials and Methods

AISI 5045 steel is widely used in mechanical engineering, including the construction of heavily loaded elements in internal combustion engines (Table 1). AISI 5045 steel ring samples with dimensions of ϕ 35 × 9 mm were prepared and were heat-treated (40 ± 2) and polished (Figure 1). The boron layer was produced by a CO_2_ laser with a power of 2 kW. Steel samples were covered with mixed amorphous boron with water glass and fused with a laser beam to protect argon. The laser treatment parameters used in the surface layer remelting process were determined by the spot size of the laser beam of 4 mm, the processing speed of 16 mm/s and the path coverage of 0.5 mm. After the laser treatment, the surface layer of steel samples was polished. The counterpart was cut from the the SAE-48 alloy journal bearing (Table 1 and Table 2) with dimensions of 15.75 × 6.35 mm (Figure 1). As a result of the laser treatment, a boron layer was produced with a content of 1–1.5% boron, a maximum thickness of 25 μm and a maximum hardness of 1800 HV (Figure 2).

Tribological comparative studies were carried out in a conformal contact with the lubrication of the friction area with the engine oil, which is widely used for lubrication of petrol and diesel engines of passenger cars and delivery vans, working with and without turbochargers that require this level of quality (Table 3).

The measurements of the friction pair co-operating under lubrication conditions were taken at the ring specimen rotational speed of 100 rpm and at a changeable unit pressure of 5, 10, 15 and 20 MPa. The measured parameters, such as friction coefficient, temperature in the friction area and linear wear as a function of changeable load were registered at real-time during the tests. The tribological tests were conducted on the T-05 block on ring tester (Figure 3) and the steel specimen was immersed in lubricants (engine oils) (Figure 3).

Scuffing test of engine oils in the sliding movement was performed utilizing the four-ball testing machine. In the tests, the balls with a diameter of 12.7 mm (0.5 in.) were used. The surface roughness, expressed as Ra parameter equaled 0.032 μm, and hardness amounted to 62 HRC ± 2. All tests during the current research were repeated three times.

## 3. Results

The measurements of surface roughness of the borided ring samples and counterparts with the bearing alloy after tests revealed significant changes, as compared to the roughness prior to the tests. The measurements of the Ra, Rz and Sm parameters of surface roughness indicated an increase of surface roughness. The measurements of Ra parameter of ring specimens showed the change of 3% in the pairs lubricated with synthetic oil and 7% in the pairs lubricated with mineral oil (Table 4).

Changes of a similar character, but on a much larger scale occurred in the case of the measurement of the Sm parameter, which increased by 62% in the friction pairs lubricated with the 15W-40 mineral oil, and by 49% in those lubricated with the 5W-40 synthetic oil. When measuring the Rz parameter, lubrication with the 15W-40 engine oil caused smaller changes in the geometric structure of the surface (19%) than in the case of using the 5W-40 engine oil (26%).

The surface roughness of the counterparts from SAE-48 bearing alloy showed more extensive changes in the measured parameters than of ring specimens with laser borided layer (Table 5). The changes of Ra, Rz and Sm parameters of the counterparts exceeded even several dozen percent. The 15W-40 mineral oil lubrication showed smaller percentage changes in surface roughness, as opposed to lubrication with the 5W-40 synthetic oil. Particularly significant changes relate to the Sm parameters, which in the friction pairs lubricated with mineral oil increased by 156% and with synthetic oil by 192%. Minor changes were observed for the Rz parameter and were 83% and 137%, respectively. The Ra parameter was increased by 44% in the pairs lubricated with mineral oil and by 78% in those lubricated with synthetic oil.

During the start-up of the friction pair, the registration of the friction coefficient allows for the determination of the energy demand necessary to start the friction pair, specification of frictional resistance in variable sliding conditions and their stabilization areas depending on the load conditions. An important parameter characterizing kinematic pairs is the maximum moment of friction during start-up of the tested friction pairs. In the tested pairs with a boron surface layer, the start-up moment was lower when lubricating the friction area with mineral oil than when lubricating it with synthetic oil (Figure 4). An important change is observed in the entire tested load range from 5 to 20 MPa. A significant difference in the start-up moment of the friction pair is observed at a load of 10 MPa, which is then 22%. Smaller difference is observed at the lowest load of 5 MPa and then the difference does not exceed 15%. At loads of 15–20 MPa, the start-up moment value is similar for both tested engine oils and does not exceed 3%.

The changes of friction resistance in the friction pairs with the laser borided surface layer in the start-up period showed a lower friction coefficient in pairs lubricated with the 5W-40 synthetic oil than in pairs lubricated with the 15W-40 mineral oil (Figure 5). For loads of 10–20 MPa, after the initial increase in the friction coefficient at the moment of start-up of the friction pair, a further increase in the friction resistance is observed along with the duration of the test. The value of the friction coefficient stabilizes only at a load of 5 MPa. In the pairs lubricated with mineral engine oil, at the load of 15–20 MPa, the course of the friction coefficient is similar and reaches similar values for all three tested loads, so that in the final stage of the test, the coefficient value is about 0.17. At the load of 15 MPa, the friction coefficient stabilizes at the level of 0.15. The variable course of the friction coefficient at the load of 5 MPa, in the initial phase of cooperation of the friction pair, leads to its stabilization after 350 s, at the level of 0.12. Under lubrication conditions with synthetic oil, at the loads of 10–20 MPa, there is a systematic increase in the value of the friction coefficient to its final value of 0.13 for the load of 10 MPa, 0,14 for 15 MPa, and over 0.15 for 20 MPa. At the lowest load of 5 MPa, the friction coefficient stabilizes below 0.09, after less than 100 s.

The friction force and temperature in the friction area allows to determine the working conditions of the friction pair, and if they show lower values, such a system ensures greater stability of operation and greater safety area in the event of an overload of the friction pair, which may lead to its seizure and destruction.

The friction force and temperature in the friction area for the pairs lubricated with the 15W-40 mineral oil are higher than in the case of the pairs lubricated with the 5W-40 synthetic oil (Figure 4). The greatest difference in the friction force occurs at the load of 5 MPa of the friction pair and it is 19% lower for the pair lubricated with the 5W-40 oil, as compared to the pair lubricated with 15W-40 oil. In the case of the remaining loads of 10–20 MPa, the difference amounts to a few percent, and in the case of the 15 MPa load, the difference is 2%, which is within the limits of the measurement method. The measured temperature in the friction area also shows a very similar course, while the differences between the tested friction pairs, depending on the engine oil used, are much smaller than those calculated in the friction force (max 4%). In the case of the15 MPa load, the temperature value for both used engine oils is identical and amounts to 98 °C (Figure 6).

The analysis of wear of the friction pair elements makes it possible to determine the durability of the friction pair, which allows to determine the period of correct operation of the entire device, as well as periods of periodic inspections and replacement of consumables.

The ring samples with the laser borided surface layer did not show the measurable linear wear, but it was possible to observe the intense wear process of the SAE-48 bearing alloy. The wear of the SAE-48 bearing alloy in the friction pair lubricated with 5W-40 synthetic oil was higher than the in the friction pair lubricated with 15W-40 mineral oil, at the load 10–20 MPa (Figure 7). The wear of the bearing alloys increases with the pressure increase in the contact area between the surface layers of both elements of the friction pairs. The difference in wear of the alloy ranges from a few percent (at the load of 20 MPa) to several percent (at the load of 10–15 MPa). At the pressure of 5 MPa, the wear of SAE-48 alloy was lower by 22%. In the friction pair lubricated with 5W-40 synthetic oil.

The behavior of a lubricant under scuffing conditions determines the areas in which the oil retains its properties, enabling the separation of the lubricated surface layers of the cooperating elements of the friction pair. During the tests on the T-02U tester, two loads can be registered: one marking the beginning of the scuffing area and the other, the so-called limiting pressure of seizure. Limiting pressure of seizure depends on the load at which the balls seize and the average value of the wear-scar diameters measured on the stationary balls.

Measurements of the seizing load of the used and unused 15W-40 and 5W-40 engine oils show a reduction in the seizure resistance of the oil used during the wear tests of the friction pair, as compared to the unused oil (Figure 8). In the tests carried out at the temperature of 40 °C, it can be observed that unused 5W-40 oil shows the highest resistance to scuffing, while at the temperature of 100 °C, both unused oils show similar values. After the cooperation, a significant reduction in resistance to scuffing of the 5W-40 oil is observed at 40 °C (13%), and for the 15W-40 oil, it decreases to 4%. At the temperature of 100 °C, the seizing load of 5W-40 oil is higher than that of 15W-40 oil, while the value of the 5W-40 oil seizing load is similar to the scuffing load of the unused oil.

The seizure load measurements for the tested oils show significant differences and it can be concluded that the 15W-40 oil shows higher load values than the 5W-40 oil (Figure 9). In the case of 15W-40 oil, the load values for used and unused oils are similar at 100 °C. On the other hand, at 40 °C the load is lower than that for the used oil. The examination of the 5W-40 oil shows a much greater decrease in the seizure load at 100 °C, which is 11%, and at 40 °C, the seizure load value is similar.

The pressure of seizure shows a significant deterioration in the performance properties of the tested engine oils, which may have been caused by contamination of the oil with wear products and a reduction in the amount of anti-seizure additives in the oil. In the conditions of friction with limited lubrication in the friction area, the anti-seizure additives contained in the engine oil are responsible for the formation of boundary layers preventing seizure, which leads to their wear. On the other hand, stabilization or slight increases in the scuffing load can be explained by the migration of boron, exhibiting lubricating properties, into the oil. Another factor is the decrease in the amount of other additives refined from engine oils, which in the new oils cause the degeneration of the lubricating properties (viscosity modifiers, corrosion or rust inhibiting, detergent additives).

An observation of the sliding surface of SAE-48 bearing alloy samples after wear tests shows non-uniformity of the friction surface. On the surface of the bearing alloy, there are areas of the surface layer with significant deformation, which results from the variable hardness of the laser treatment layer and the lines formed on it, resulting from the arrangement of overlapping traces of the laser beam (Figure 10). In these areas, significant damage to the sliding layer of the bearing alloy can also be observed—cracks and flaking. The maps of the elements in the designated friction areas show significant differences in the distribution of the selected elements Cu, Pb, S, Mo and B (Figure 11).

An observation of the base elements of the SAE-48 bearing alloy shows a uniform distribution of Cu, while in the case of Pb, significant differences can be observed, which may result from the migration of lead from the heavily loaded areas to the boundary layers formed in the cooperation area. The analysis of the distribution of S and Mo elements, which are components of the engine oils, also shows their uneven distribution. Intensive wear processes occur in the areas with low presence of the S and Mo elements, which impede the formation of permanent boundary layers containing these elements. The maps of boron distribution shows single atoms in the friction area, which may indicate that in this type of friction pair composition boron transfer is negligible or this element is washed away by the lubricant.

## 4. Discussion

Lubrication of friction pairs with laser remelting structural elements significantly shapes the processes of friction and wear by formation of boundary layers, as a result of the reaction of the surface layers of these elements and the lubricant. The observation of the geometric structure of the surface layers shows a significant reorganization, the effect of which is an increase in the surface roughness of both layers, especially the surface of counter-samples of the SAE-48 bearing alloy [33].

Changes in the Ra parameter show a comprehensive picture of the surface, and the changes in the surface roughness result mainly from the adaptation of both surface layers of the cooperating elements to the specific load conditions of the friction pair. Changes in this parameter significantly affect the load-bearing capacity of the operating surface layers formed in the process of friction. The+ Rz and Sm parameters are parameters that allow the assessment of the local surface unevenness field, and significant changes in their size allow for the identification of changes that arise as a result of drawing, grooves, micro-cutting to protruding surface irregularities or hard wear products occurring in the area of friction or grafting processes occurring as a result of the disappearance of the lubricating layer. These changes result from the processes of rapid destruction of the existing lubricating layers or difficulties in shaping them as a result of friction processes and direct contact of the cooperating layers of the friction pair elements [34]. The unfavorable effect of the laser borided layer may be the result of the variable hardness of these surface layers, which is the result of a specific method of its shaping, by overlapping successive laser beam (Figure 10). The observation of the course of the friction coefficient shows that low unit pressures influence the shaping of favorable lubrication conditions, and the system changes the existing geometric structures of both elements into a composition ensuring the most favorable cooperation conditions. The resulting structure provides the given pair with optimal functionality, caused by the creation of stable operational surface layers on both elements of the friction pair [35,36].

The friction and wear processes taking place in the contact area of the sliding pairs depend on unit pressure between cooperating surface layers, microstructural changes of surface layer, chemical reactions between materials, material transformation and material properties variation in surface layers [33,37,38]. The wear of bearing alloy is mainly caused by the hard areas in the surface layer of the second material and hard wear particles, which leads to the interaction between the two surface layers and a more intense abrasion of the softer material [37,38]. Also, an increase in roughness of the harder material results in an increase in the wear particles detached from the softer material [33]. The hard wear products created in the friction process induce chipping, slicing and grinding, which intensify the wear process [36,39].

During operation, engine oil changes its properties and accumulates impurities, which worsens the lubrication conditions and increases the intensity of wear processes. The lubrication process can remove wear products from the friction area. However, as a result of tribochemical reactions, corrosion processes, cavitation wear, or fatigue wear may occur, the latter caused by dynamic changes in the pressure in the friction area. The assessment of the phenomena is complex and only on the basis of the surface observation of the elements can it be concluded that the oil affects the processes occurring in the contact area.

## 5. Conclusions

Based on the experimental test, the following conclusions can be drawn:The surface roughness of the SAE-48 bearing alloy and laser borided ring specimen shows tendencies to increase and a greater change was observed in the pairs lubricated with 5W-40 engine oil.Friction force in the pair lubricated with the 15W-40 engine oil is considerably higher than in the pair lubricated with the 5W-40 oil, and temperature in the friction area for both tested oils is of a comparable value level.The application of the 5W-40 oil in the sliding pair with the laser borided surface layer causes a significant increase of wear of the SAE-48 bearing alloy, together with the increase of unit load in the friction area.The processes of wear lead to the destruction of engine oil and the deterioration of their resistance to scuffing, especially during the operation under low temperature.Lubrication of the friction pair with laser borided surface layer with the 15W-40 oil is more advantageous because it causes smaller changes in the geometrical structure of the cooperating surface layers, lower friction pair starting resistance and lower wear at a comparable temperature in the friction area.

## Figures and Tables

**Figure 1 materials-13-05810-f001:**
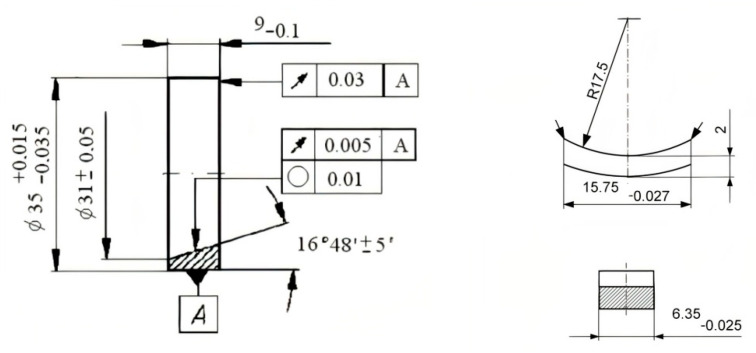
Dimensions of elements of the tested pair (mm).

**Figure 2 materials-13-05810-f002:**
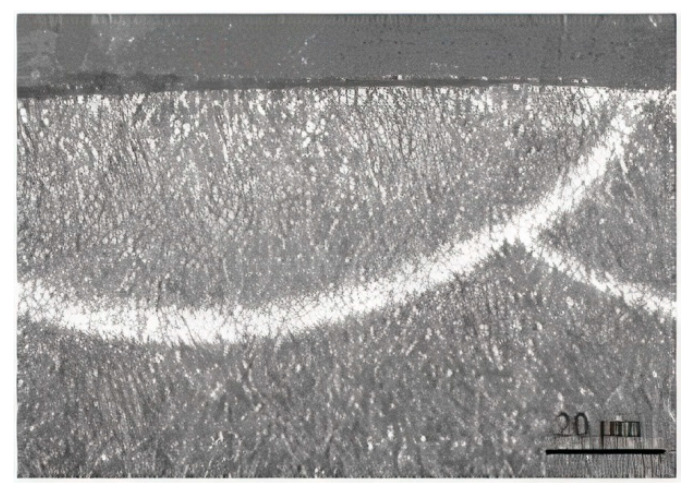
Microstructure of the steel specimen with laser borided surface layer.

**Figure 3 materials-13-05810-f003:**
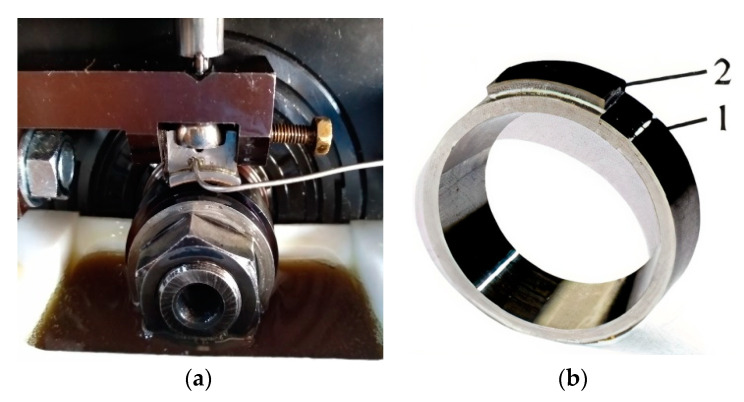
The friction pair in block on ring tester (**a**) and composition of friction pair: 1—ring specimen, 2—counterpart (**b**).

**Figure 4 materials-13-05810-f004:**
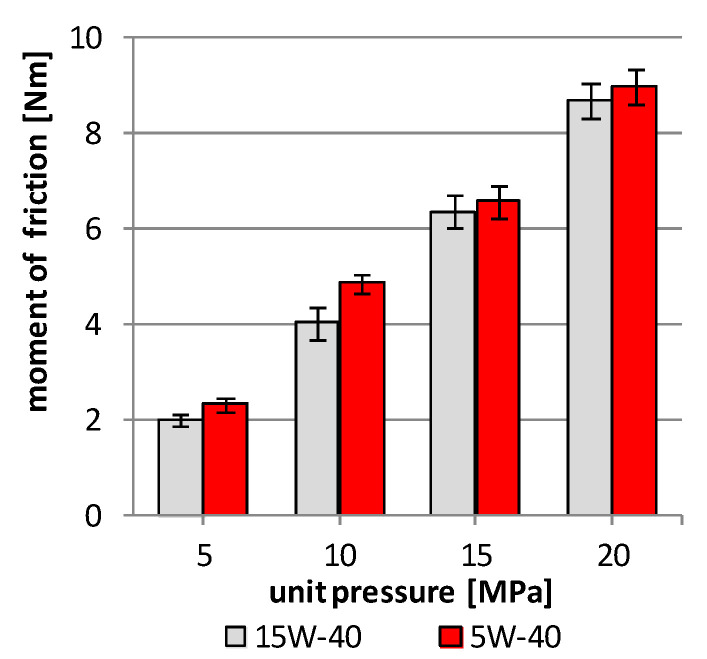
Moment of friction during start-up of the pair vs unit pressure when lubricating with 15W-40 mineral engine oil and 5W-40 synthetic engine oil.

**Figure 5 materials-13-05810-f005:**
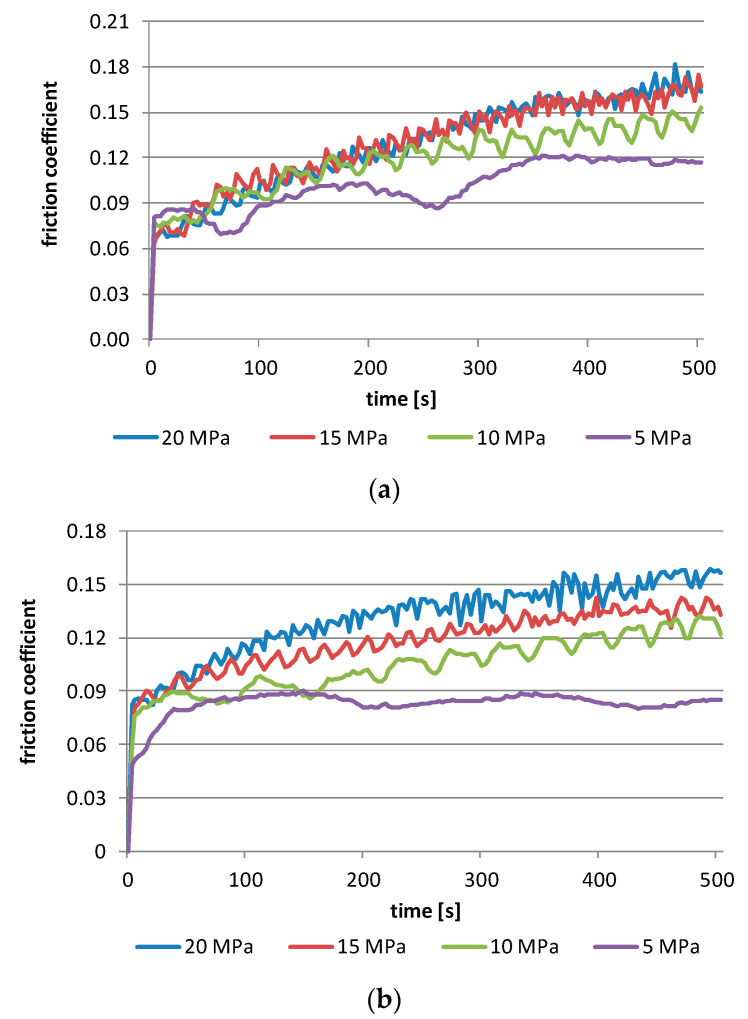
Friction coefficient in a sliding couple with laser borided surface layer vs rotation speed, lubricated with (**a**) 15W-40 mineral engine oil, (**b**) 5W-40 synthetic engine oil.

**Figure 6 materials-13-05810-f006:**
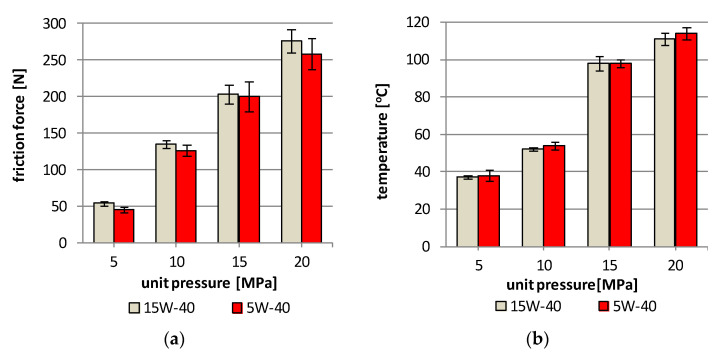
Friction forces (**a**) and temperature in friction area (**b**) vs unit pressure (at 100 rpm and after 500 s).

**Figure 7 materials-13-05810-f007:**
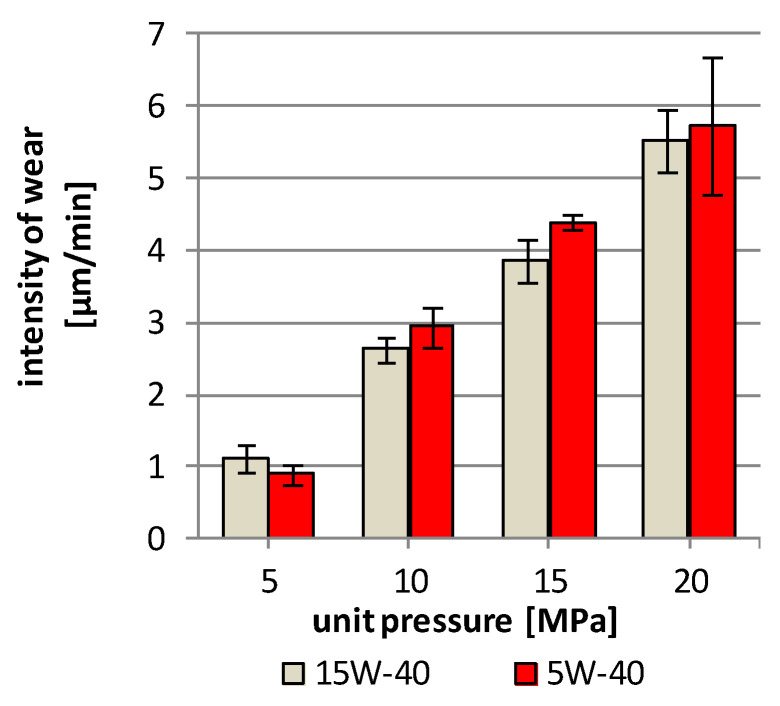
Intensity of wear of SAE-48 bearing alloy during lubrication by engine oils (at 100 rpm).

**Figure 8 materials-13-05810-f008:**
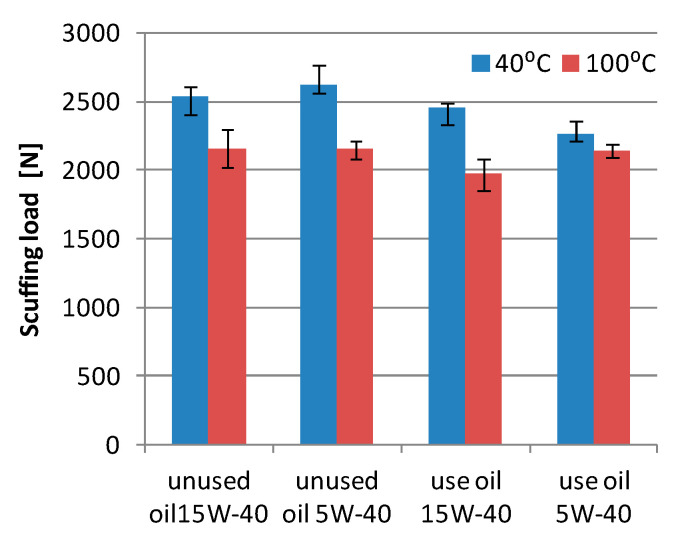
Scuffing load of the unused and used oil; 15W-40 and 5W-40.

**Figure 9 materials-13-05810-f009:**
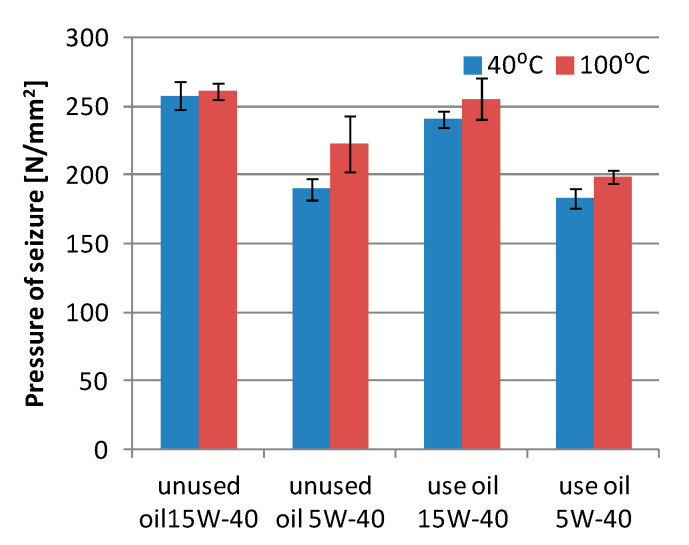
Pressure of seizure of the unused and used oil; 15W-40 and 5W-40.

**Figure 10 materials-13-05810-f010:**
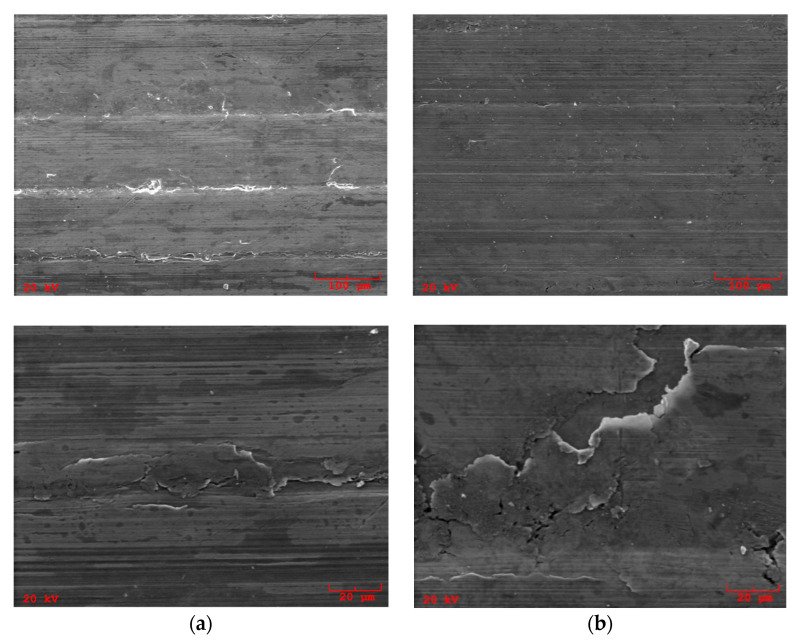
Micrographs of SAE-48 bearing after tests; (**a**) 15W-40 engine oil; (**b**) 5W-40 engine oil.

**Figure 11 materials-13-05810-f011:**
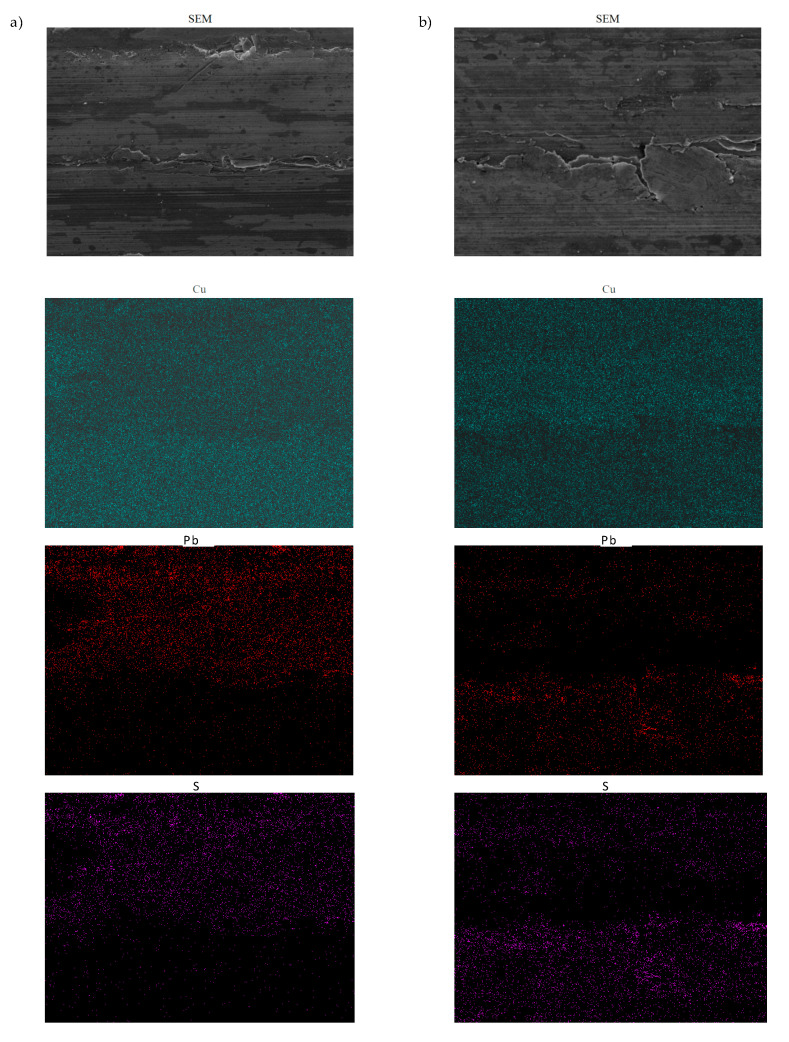
Maps of the distribution of elements in the surface layer SAE-48 bearing alloy; (**a**) 15W-40 engine oil, (**b**) 5W-40 engine oil.

**Table 1 materials-13-05810-t001:** Chemical composition of specimens (wt.%).

Material	C	Cr	Mn	Si	Fe	Pb	Cu
AISI 5045	0.46	0.5	0.65	<0.4	Balance	-	-
SAE-48	-	-	-		-	26–33	Balance

**Table 2 materials-13-05810-t002:** Mechanical properties of SAE-48 bearing alloy (CuPb30).

Operating temperature Tmax (°C)	170
Load Pmax (N/mm^2^)	140
Tensile Strength (N/mm^2^)	200
Sliding Speed V-oil Lubricated (m/s)	8
Alloy Hardness (HB)	30–45

**Table 3 materials-13-05810-t003:** Characteristics of engine oils.

Parameter	5W-0 Synthetic Oil	15W-40 Mineral Oil
Kinematic Viscosity at 100 °C	13.7 mm^2^/s	14.5 mm^2^/s
Viscosity Index	178	133
HTHS Dynamic Viscosity at 150 °C	3.6 mPa·s	3.8 mPa·s
Specification	API-SL/SJ/CF/CD ACEA-A3-98/B3-98/B4-98 ACEA-A3/B4-04	API SJ/CF ACEA A3-02/B3-98

**Table 4 materials-13-05810-t004:** Roughness of steel ring specimens after the test under pressure of 20 MPa (DIN 4768, ISO 4287).

Engine Oil	5W-40 Synthetic Oil		15W-40 Mineral Oil	
Parameter	Value [μm]	Change [%]	Value [μm]	Change [%]
Ra	0.31	3	0.32	7
Rz DIN	3.4	26	3.2	19
Sm	82	49	89	62

**Table 5 materials-13-05810-t005:** Roughness of counterparts from SAE-48 after the test under pressure 20 MPa (DIN 4768, ISO 4287).

Engine Oil	5W-40 Synthetic Oil		15W-40 Mineral Oil	
Parameter	Value [μm]	Change [%]	Value [μm]	Change [%]
Ra	0.89	78	0.72	44
Rz DIN	7.1	137	5.5	83
Sm	140	192	123	156

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
