# Peer review of "Experimental Analysis of Tribological Processes in Friction Pairs with Laser Borided Elements Lubricated with Engine Oils"

_materials, 2020, doi:10.3390/ma13245810_

Round 1

Reviewer 1 Report

The paper looks good overall. A lot of experiments were carried out. But I feel there is still room for this paper to be improved. My comments are listed as follows,

1 Are the engine oils in the circulating condition during the wear test or the wear pair was just immersed inside lubricants(oils)?

2 It would be better if the authors could provide one figure describing the microstructure of the component after the laser treatment.

3 What is the surface condition of the laser treated sample for the wear test? Grinding or as-treated.

4 What was the test pressure for the roughness data reported in Table 4.

5 Ra, Rz and Sm with different oils were described in details. Which value would be the best one/popular/reliable to evaluate the effect of the oil on the wear performance?

6 Table 5 is missing.

7 A lot of wear parameters such as friction coefficient, intensity of wear, friction force, moment of friction, etc. were obtained from the wear test, which made this paper very informative. But why these parameters were important and deserve the authors spent effort on reporting? It is better to add a few sentences to describe the importance of these parameters before starting the analyzing the experimental data.

8 Why the authors didn’t analyze the wear scar of the laser treated component after the wear test?

9 Which oil is better? Or Which oil is better under specific test condition? These important information should be made in the conclusions part.

Author Response

Thank you for your comments and suggestions that allowed us to greatly improve the quality of the manuscript. We agree with all your comments.

Your comments are in bold text and our responses in plain italics.

1 Are the engine oils in the circulating condition during the wear test or the wear pair was just immersed inside lubricants(oils)?  

The steel specimen was immersed in lubricants  (line 156 and Figure 2)

2 It would be better if the authors could provide one figure describing the microstructure of the component after the laser treatment.

The microstructure of steel specimen with laser borided surface layer (Figure3)

3 What is the surface condition of the laser treated sample for the wear test? Grinding or as-treated.  

The surface of the laser- treated sample was polished (line 113)

4 What was the test pressure for the roughness data reported in Table 4.

The pressure was 20 MPa.

5 Ra, Rz and Sm with different oils were described in details. Which value would be the best one/popular/reliable to evaluate the effect of the oil on the wear performance?

Changes in the Ra parameter show a comprehensive picture of the surface, and changes in the surface roughness result mainly from the adaptation of both surface layers of the cooperating elements to the specific load conditions of the friction pair, and changes in this parameter significantly affect the load-bearing capacity of the operating surface layers formed in the process of friction. The Rz and Sm parameters are parameters that allow the assessment of the local surface unevenness field, and significant changes in their size allow to identify changes that arise as a result of drawing, chasing or cutting processes by hard wear products occurring in the area of friction or grafting processes occurring as a result of damage to the lubricant boundary layer.

6 Table 5 is missing.

The table has been inserted.

7 A lot of wear parameters such as friction coefficient, intensity of wear, friction force, moment of friction, etc. were obtained from the wear test, which made this paper very informative. But why these parameters were important and deserve the authors spent effort on reporting? It is better to add a few sentences to describe the importance of these parameters before starting the analyzing the experimental data.

A few sentences were added to each chapter (lines 212, 246, 277)

8 Why the authors didn’t analyze the wear scar of the laser treated component after the wear test?

The wear scars of the laser treated component are under development and show the processes of transferring materials from the counterpart (bearing alloy), and, additionally, there are elements and chemical compounds that have appeared or separated as a result of chemical reactions of oils with other elements of the pair- which will be the subject of a separate study.

9 Which oil is better? Or Which oil is better under specific test condition? These important information should be made in the conclusions part.

Conclusion nr. 5

Lubrication of the friction pair with laser borided surface layer with the 15W-40 oil is more advantageous because it causes smaller changes in the geometrical structure of the cooperating surface layers, lower friction pair starting resistance and lower wear at a comparable temperature in the friction area.

Reviewer 2 Report

Here are my comments and questions about the paper "Influence of engine oil viscosity grades on tribological processes in friction pairs with laser borided elements":

What are the definitions of Ra, Rz, Sm in Table 4?

Fig. 5 is not cited in the paper.

How is the intensity of wear defined and measured?

What does "linear areas of material deformation" mean?

In Fig.8 & Fig. 9, figures of 15W-40 engine oil and the 5W-40 engine oil cases are provided. I wonder if the authors can provide some discussions and comparisons of these two cases based on these figures?

The introduction needs improvement. The authors need to clearly describe the scientific significance of the present study.

The authors should consider adding a figure to clearly illustrate the dimension of the specimen, and the process of the "friction and wear" experimental tests.

One main concern I have for this paper is its lack of in-depth theoretical analysis of the experimental results. Is it possible to develop or apply any mechanics theories or numerical modeling tools to analyze the friction and wear mechanisms underlying the observed phenomena?  

Author Response

Thank you for your comments and suggestions that allowed us to greatly improve the quality of the manuscript. We agree with all your comments.

Your comments are in bold text and our responses in plain italics.

What are the definitions of Ra, Rz, Sm in Table 4? 

According to DIN 4768 and  ISO 4287

Fig. 5 is not cited in the paper.

A mistake in the numbering, figure 4 was given twice, the correct numbering was entered.

How is the intensity of wear defined and measured?

Intensity of wear is the quotient of the linear wear measured with the inductive sensor and the test duration.

What does "linear areas of material deformation" mean?

It was mistake

In Fig.8 & Fig. 9, figures of 15W-40 engine oil and the 5W-40 engine oil cases are provided. I wonder if the authors can provide some discussions and comparisons of these two cases based on these figures?

Discussions are provided at the end of this section (line 320)

The introduction needs improvement. The authors need to clearly describe the scientific significance of the present study.

The scientific significance of the present study is presented at the end of the introduction.

The authors should consider adding a figure to clearly illustrate the dimension of the specimen, and the process of the "friction and wear" experimental tests.

Figure 1 presents the dimension of the specimen and figure 2 presents the construction of the friction pair in T-05 block on ring tester and the picture of the friction pair itself.

One main concern I have for this paper is its lack of in-depth theoretical analysis of the experimental results. Is it possible to develop or apply any mechanics theories or numerical modeling tools to analyze the friction and wear mechanisms underlying the observed phenomena?

The test results were analyzed statistically. The theoretical analysis is possible using FEM, which is often used for this type of friction junction design. However, due to comparative studies, no modeling of the friction process was carried out, which is also due to the fact that the research program for this friction pair composition was not finished.

Reviewer 3 Report

The article presents a very current problematics. As for the presentation of research and interpretation of results, I have no comments. I have only a few comments of a formal nature.

  • I think the reference to Figure 2 is missing in the text.
  • line 181 - 98oC
  • line 188 - there is a reference to Figure 4, but I think there should be a reference to Figure 5
  • line 218 - differences in the distribution of the selected elements Bu Cu, Pb, S, Mo and B (Figure 9). 
  • the text on the lines 249 -258 is the same as the text on the lines 259-268

Author Response

Thank you for your comments and suggestions that allowed us to greatly improve the quality of the manuscript. We agree with all your comments.

Your comments are in bold text and our responses in plain italics.

  • I think the reference to Figure 2 is missing in the text.  

It was added

  • line 181 - 98oC

It has been corrected -line 273

  • line 188 - there is a reference to Figure 4, but I think there should be a reference to Figure 5 

It has been changed (now is figure 7-line 283).

  • line 218 - differences in the distribution of the selected elements Bu Cu, Pb, S, Mo and B (Figure 9).

 It has been corrected (line 358)

  • the text on the lines 249 -258 is the same as the text on the lines 259-268

It was deleted.

Reviewer 4 Report

The current work studies the "influence of  engine oil viscosity grades on tribological processes in friction pairs with laser borided elements".

The authors perform friction tests using two oils: an synthetic oil 5W40 and a mineral one 15W40.

The authors draw conclusions based on the viscosity only, however the base oils (mineral and synthetic) as well as the additive packages in both oils could also generate the obsrved difference in behaviour.

An example, the higher viscoity oil gives the lower friction (fig 2), the synthetic oil gives the lower friction (fig 3), the first explanation is likely viscosity based, the second one is surely additive based.

If the authors want to investigate the influence of viscosity on friction and wear, they are advised to stick to one base oil type (synthetic OR mineral) and add the same type additive package to both oils. That way measured differences can be attributed to viscosity.

As it is, it is impossible to decide if the observed difference stems from the different viscosities, base oils or additives.

For these reasons the reviewer cannot recommend the current work for publication

Author Response

Thank you for your comments and suggestions that allowed us to greatly improve the quality of the manuscript. We agree with all your comments.

Your comments are in bold text and our responses in plain italics.

The authors perform friction tests using two oils: an synthetic oil 5W40 and a mineral one 15W40.

The authors draw conclusions based on the viscosity only, however the base oils (mineral and synthetic) as well as the additive packages in both oils could also generate the obsrved difference in behaviour.

An example, the higher viscoity oil gives the lower friction (fig 2), the synthetic oil gives the lower friction (fig 3), the first explanation is likely viscosity based, the second one is surely additive based.

If the authors want to investigate the influence of viscosity on friction and wear, they are advised to stick to one base oil type (synthetic OR mineral) and add the same type additive package to both oils. That way measured differences can be attributed to viscosity.

As it is, it is impossible to decide if the observed difference stems from the different viscosities, base oils or additives.

For these reasons the reviewer cannot recommend the current work for publication

I fully agree with the oil research concept presented in the review.
However, the concept of the publication includes comparative research aimed at determining the behavior of the real friction pair in the selected combustion engines. The only modification in a given friction pair composition replacing the surface treatment of the journal from heat treatment to laser boriding. In real friction pairs, both types of engine oils are used for lubrication, which means that research on the base oil and the use of additives is not justified in this case, because it does not reflect the real composition of the tested pair.
The application of the procedure algorithm presented in the review is possible in the conditions where we do not refer to the real friction nodes in internal combustion engines operating under lubrication conditions with dedicated engine oils

Round 2

Reviewer 1 Report

My comments and questions had been addressed carefully by the authors. I think the revised paper is way much better than the previous version. I suggest accept the current version.

Author Response

We are very grateful to you for taking the time to write a positive opinion - thank you very much.

Reviewer 4 Report

The reviewer considers that the authors have NOT answered my main question, different viscosity versus different additives,

as such the reviewer cannot recommend the current work for publication

Author Response

Based on the conducted research, it is not possible to determine the effect of oil viscosity and oil additives on the friction processes. Under the conditions of limited lubrication that occurred during the tests, there is no lubricating film (no hydrodynamic lubrication), which causes that the viscosity of the lubricant has little (or no) impact on the processes of boundary layer formation. The main factor influencing the friction processes is then the characteristics of the cooperating surface layers and the chemical properties of the lubricant. Lubricant layers (boundary layers) are mainly formed by tribochemical reactions, and their composition is the result of the composition of the materials in the friction pair (including additives in the engine oil) and the friction conditions. Boundary layers must adhere strongly to the surface of the solid to prevent deterioration or scuffing. They have high cohesive strength to resist shear and good surface reactivity so that the active ingredients of the lubricant regenerate the new boundary layer for the next sliding cycle. Due to the fact that commercial oils were used in the research, the authors are not able to determine the composition of enriched additives and their influence on the friction processes - it was not the aim of the research either. The research objective was to determine the influence of the selected classes of engine oils on friction and wear processes in friction pairs containing elements with a laser borided surface layer under conditions of limited lubrication.